# From Conventional Disinfection to Antibiotic Resistance Control—Status of the Use of Chlorine and UV Irradiation during Wastewater Treatment

**DOI:** 10.3390/ijerph19031636

**Published:** 2022-01-31

**Authors:** Muhammad Umar

**Affiliations:** Norwegian Institute for Water Research (NIVA), Økernveien 94, 0579 Oslo, Norway; muhammad.umar@niva.no

**Keywords:** wastewater treatment, antibiotic resistance genes, DNA, UV treatment, advanced oxidation processes

## Abstract

Extensive use of antibiotics for humans and livestock has led to an enhanced level of antibiotic resistance in the environment. Municipal wastewater treatment plants are regarded as one of the main sources of antibiotic-resistant bacteria (ARB) and antibiotic resistance genes (ARGs) in the aquatic environment. A significant amount of research has been carried out to understand the microbiological quality of wastewater with respect to its antibiotic resistance potential over the past several years. UV disinfection has primarily been used to achieve disinfection, including damaging DNA, but there has been an increasing use of chlorine and H_2_O_2_-based AOPs for targeting genes, including ARGs, considering the higher energy demands related to the greater UV fluences needed to achieve efficient DNA damage. This review focuses on some of the most investigated processes, including UV photolysis and chlorine in both individual and combined approaches and UV advanced oxidation processes (AOPs) using H_2_O_2_. Since these approaches have practical disinfection and wastewater treatment applications globally, the processes are reviewed from the perspective of extending their scope to DNA damage/ARG inactivation in full-scale wastewater treatment. The fate of ARGs during existing wastewater treatment processes and how it changes with existing treatment processes is reviewed with a view to highlighting the research needs in relation to selected processes for addressing future disinfection challenges.

## 1. Introduction

Wastewater reuse has traditionally been considered safe after treatment by a combination of physicochemical and biological treatments that target the removal of organic matter, chemical contaminants, and microorganisms. Consequently, the benchmarks for assessing treatment performance that are currently incorporated into the regulations generally include total suspended solids, chemical oxygen demand, biochemical oxygen demand, ammonia, nitrate, total phosphorus, and inactivation of microbes such as fecal coliforms or *Escherichia coli* (*E. coli*) [1]. However, microbial inactivation alone is not sufficient to ensure the safety of treated wastewater since genes may still be present even after disinfection [2]. This requires a shift from conventional microbiological inactivation to sufficient DNA damage to minimize the spread and development of antibiotic resistance. Importantly, the current disinfection processes are not designed to damage genes, which makes antibiotic resistance genes (ARGs) a serious concern since they enable microorganisms to become resistant to antibiotics.

Municipal wastewater is regarded as one of the main sources of antibiotic-resistant bacteria (ARB) and ARGs in the aquatic environment. In fact, wastewater treatment plants (WWTPs) are considered a major “hotspot” of ARB and ARGs since these have been frequently found in WWTP effluents [3]. This is primarily because the current WWTPs are not equipped with appropriate technologies for their mitigation/inactivation prior to or upon effluent reuse or discharge. It is well known that the final effluents (particularly biologically treated effluents) release a high number of bacteria and genes that may be resistant to antibiotics [4]. Existing literature suggests that the high growth rates and microbial densities that are fundamental to conventional biological treatments at WWTPs, along with the presence of residual antibiotics, could create a highly suitable environment for promoting ARG transfer and multiantibiotic resistance among bacteria [5]. Biological processes create an environment that is potentially suitable for resistance development and spread due to the continuous interaction of bacteria with antibiotics at sub-inhibitory concentrations. For example, the Class 1 integron gene *intI*1 has been reported to increase in WWTP effluents, indicating the fitness gain of bacteria harboring this mobile element [6]. Moreover, the potential for co-selection of ARGs is increased when the bacterial communities are exposed to a chemical stress (heavy metals, antibiotics, or both) in a continuously changing WWTP environment [7]. Various investigations have shown a correlation between the presence of genes encoding for resistances against different metals and of ARGs in plasmids and integrons associated with contaminated soils and WWTPs [8,9,10].

Such concerns may impede efforts to provide safe drinking water as well as the discharge and reuse of the treated wastewater. Since there is an increasing interest in and demand for the safe discharge and/or reuse of wastewater for applications such as irrigation, aquaculture, aquifer recharge, indirect potable reuse, and even direct potable reuse, an understanding and mitigation of the risks arising from ARB and associated ARGs are very important. It is particularly relevant considering that wastewater reuse (especially for agricultural purposes) and biosolids that are generated during treatment are considered important means of AMR spread.

UV disinfection as an alternative to chlorine is of particular interest since UV light is directly absorbed by DNA and therefore has the potential to inactivate ARGs and limit their release and transfer to bacteria. Although the research on the efficiency of UV disinfection in damaging ARGs is recent, it is one of the most promising technologies to address this emerging issue [2]. However, the process is inherently expensive due to the high energy requirements. Moreover, UV photolysis alone may not be as efficient for damaging ARGs since the extent of DNA damage is dependent on the UV fluence, which is generally higher for achieving DNA damage than microbiological inactivation, cell structure, and water matrix [11]. For real wastewaters, the process can be enhanced by using oxidants such as H_2_O_2_ and Cl_2_ in what are termed advanced oxidation processes (AOPs). Direct UV photolysis, UV/H_2_O_2_, and UV/Cl_2_ are focused on in this article considering their practical applications and greater future potential compared with other disinfectants and AOPs for ARG inactivation. Since these processes and/or their combinations are widely used in practical water and wastewater treatment for a range of contaminants, they stand out as some of the best potential technologies for inactivation of ARGs on a large scale. Considering that an AOP produces radicals that can break down organic matter while simultaneously inactivating bacteria and ARGs, this process can be useful for both water treatment and wastewater reclamation and reuse [12]. Several recent investigations have shown these processes to be effective but to a varying degree depending on a range of process and operational parameters. This article specifically focuses on the potential of these processes for practical application at full scale for the purpose of inactivating ARGs considering the findings recently reported on the topic. This review specifically focuses on selected technologies to: (1) summarize and review the progress over the past several years; and (2) identify and discuss areas of importance in relation to the state of the art with particular reference to challenges needed to be addressed for full-scale applications. Knowledge gaps related to the significance of target ARGs and the effectiveness of UV and UV-based AOPs in various water matrices with respect to changes in the resistance shift and impact of co-existing contaminants are identified and reviewed in the context of knowledge needs for future applications of these processes.

## 2. Antibiotic Resistance—Mechanisms of Resistance Spread and Wastewater Treatment

### 2.1. Mechanism of Spread and Selection of ARGs

Antibiotic resistance is the ability of bacteria to survive and potentially thrive in the presence of antibiotics [12]. Extensive use of antibiotics in human and animal health, and their use for the promotion of growth in animals, has accelerated the process of microbial resistance. Resistance to antibiotics is encoded in ARGs, enabling bacteria to fight antibiotics through various mechanisms. The two mechanisms responsible for transfer of antimicrobial resistance include: vertical gene transmission (VGT), which involves inheriting the genetic information from the parent cells; and horizontal gene transfer (HGT), in which a non-resistant bacterium becomes resistant by gaining the resistance genes other than from its parent cell [2]. Antibiotic resistance spread mechanisms are shown in Figure 1. HGT occurs through conjugation, transduction, and transformation. Notably, VGT and conjugation occur when the gene-carrying bacterium is viable whereas successful transduction occurs when the virus carrying the gene is also infective [2]. Transformation, however, can occur without a viable or infective donor microorganism since bacteria can obtain ARGs from cell-free DNA [13]. Inactivation of microorganisms alone, without effective DNA damage, as a primary objective, is therefore not sufficient because it does not guarantee control over the spread of antimicrobial resistance [2].

ARGs are therefore considered a contaminant of emerging concern (CEC) considering that they present “new” environmental and public health concerns. Contrary to chemical CECs, which have been widely focused on over the past several years, perspectives on antibiotic resistance in general and with respect to water reuse in particular are unclear. For example, knowledge on the fate and concentrations of chemical CECs is well established and suitable approaches and technologies have been developed for mitigating their impact. However, very little is known about ARGs in terms of the safe level, the types of genes to prioritize in the broader perspective, and methods for and approaches to mitigating resistance risk. It is attributed predominantly to the lack of knowledge and understanding of the complexity of and the multiple dimensions to the issue of microbial resistance. Although the existing water reuse regulations and guidelines do not address concerns related to AMR, this trend is poised to change with increasing focus on effective treatments for the safe reuse of the treated wastewater. Currently, most wastewater treatment processes lack the ability to sufficiently destroy nucleic acids and thus reduce the downstream spread of resistance. In fact, some treatment processes, including commonly used processes such as disinfection (e.g., chlorination), have been reported to increase the level of resistance through a selective increase in certain ARBs [15]. For example, chlorine has been known to select for certain ARB and ARGs under conditions typically used in water treatment and elevate the average resistance of ARB [15], which may lead to the selection of organisms resistant to higher concentrations of antibiotics.

Therefore, the optimization and incorporation of alternative, and advanced, wastewater processes are important steps in the mitigation of the antibiotic resistance spread. This is even more important considering that the treatments designed for the removal of conventional contaminants could potentially be contributing to an increased risk of ARB and ARG selection and spread. Consequently, inefficient removal of ARB and ARGs could further compound the problem of antibiotic resistance due to a selective increase in more resistant bacteria. It therefore depends on the type of treatment as well as the type of drug family present in the wastewater [16]. For example, the presence of ampicillin, sulfamethoxazole, ciprofloxacin, and tetracycline-resistant *E. coli* during different treatment stages of a WWTP (raw sewage, post-secondary, post-UV, and post-chlorination) was investigated [16]. Resistance to three or more antibiotics (multidrug resistance) was reported in 21% of the *E. coli* isolates. The minimum inhibitory concentration (MIC) values suggested that the resistance to ampicillin was most common among the multidrug-resistant *E. coli* with four of the isolates showing an ampicillin MIC > 256 µg/mL. Three of these high MIC values were after UV and chlorination treatment of the finished water. Two *E. coli* isolates were also found to be resistant to tetracycline with a MIC > 256 µg/mL [16]. Ampicillin, ciprofloxacin, and trimethoprim–sulfamethoxazole multidrug resistance was observed in 15 *E. coli* isolates (75%), where one isolate was resistant to all four antibiotics. Overall, the authors concluded that tetracycline resistance was the least among the isolates whereas ampicillin-resistant *E. coli* was the most predominant (63% of the total *E. coli* population) after UV and chlorine disinfection. It was further noted that the ABR *E. coli* concentrations in the effluent were higher after the rainfall event.

These findings are supported by another investigation [17] that performed a metagenomic study on sewage-derived microorganisms from WWTP influent, showing that the diversity in the microbial community increased after the rainfall. It was attributed to low disinfection efficiency due to increased flow and a reduced retention time. It shows that the role of environmental conditions is important in addition to the type of treatment applied. Not much work, however, has been done to determine the changes in the concentration of ARB and ARGs due to rainfall, requiring more investigations to understand the impact of rainfall events. Nonetheless, the effect of different treatments on the fate of ARGs has been increasingly evaluated and is the focus of the following section.

### 2.2. Fate of ARGs in Wastewater Treatment

A number of investigations have been performed identifying the fate of ARGs in conventional wastewater treatment scenarios. For example, quantitative and qualitative changes in the level of various ARGs (*tet*X, *tet*M, *tet*A, *sul*1, *sul*2, *erm*B, *qnr*D, and *bla*_TEM_) were evaluated in two municipal WWTPs receiving influent mixed with pretreated livestock water (WWTP1) or industrial wastewater (WWTP2) [18]. The level of ARGs shifted markedly during different treatment stages with significant differences between the two WWTPs as well as ARGs. For example, the total number of ARGs in final effluent showed an increase of 10% for WWTP1 whereas a decrease of 75% was observed for WWTP2. The differences at the individual gene level were also reported to be significant. For example, *sul*, *qnr*D, and *bla*_TEM_ increased markedly during the treatment processes only in WWTP1 (receiving pretreated livestock water), whereas no such trend was observed for WWTP2. Most of the ARGs showed an increase ranging between 113% and 564%, except for *tet* and *erm* for WWTP1. This trend was quite different to WWTP2, for which most of the ARGs showed a decrease of 22–92% except for *tet*, which increased by 29%. These differences could be attributed to the different qualities of the influent wastewaters for both the WWTPs as well as the level of antibiotics. Importantly, the greatest shift in the ARG abundance was noted during coagulation, secondary sedimentation, and biological treatments, emphasizing the need to closely monitor these processes for changes in the composition of ARGs.

In another study, total bacterial abundance (estimated from copy numbers of the bacterial 16S rRNA gene) at five different municipal WWTPs did not reduce after wastewater treatment [19]. While the relative abundance of ARGs was generally similar before and after treatment, the *bla*_CTX-M_, *bla*_TEM_, and *qnr*_S_ genes were higher in the effluent of one of the WWTPs. Notably, this particular WWTP received untreated hospital wastewater in addition to domestic and industrial wastewater. Overall, their results agreed with some previous findings that the ARGs and the *intl*1 gene are inefficiently removed during conventional wastewater treatment [20,21].

Some studies have looked at the interaction of various classes of ARGs from a co-selection perspective. Correlation between six ARGs (*tet*A, *sul1*I, *bla*_TEM_, *bla*_CTXM_, *erm*B, and *qnr*S), two heavy metal resistance genes (HMRGs; *czc*A, *ars*B), and the mobile genetic element class I integron was investigated for three WWTPs during different treatment steps [22]. Class 1 integrons are closely correlated to co-selection mechanisms and are often associated with gene cassettes having both HMRGs and ARGs [10,22]. It has been shown that the bacterial strains having class I integrons have a selective advantage compared with the rest of the bacterial community [23]. An increase in the level of class I integron genes in the effluent therefore indicates the fitness gain of bacteria carrying this mobile genetic element. The authors noted two well-defined groups, which included (1) *tet*A, *erm*B, and *qnr*S and (2) *sul1*I, *czc*A, *ars*B, and *int*1 [22]. Overall, a strong correlation between *sul1*I, HMRGs, and *int*1 was noted. Furthermore, the authors concluded that *czc*A and *sul1*I could be used as model genes for investigating co-selection in WWTPs. Both *tet*A and *qnr*S are harbored by Gram-negative bacteria, whereas *erm*B is associated with Gram-positive bacteria. Therefore, it was unlikely that the co-presence and similar distribution of these genes indicate co-occurrence in the same cells. It is plausible to hypothesize that these clustered genes were present within the microbial community of WWTPs comprised of both Gram-negative and Gram-positive bacteria. Another study found *qnr*S-like genes in Gram-positive bacteria, i.e., *Enterococcus faecalis* and *Enterococcus faecium,* which could lead to the potential co-presence of *qnr*S-like genes and *erm*B in the same cells [24]. For the second cluster, a very strong correlation (Pearson’s; *p* < 0.0001) between *czc*A and *sul1*I and *ars*B and *int*1 was found, whereas the correlation between *sul1*I and *ars*B and *czc* and *int*I was weak yet significant (*p* < 0.01).

These findings indicate the significance of determining the mechanisms of co-selection to better understand the correlations between different genes and mobile genetic elements during different treatment stages in WWTPs. Future investigations should however be carried out since the above-described study is one of the very few to have reported this correlation. However, it can be hypothesized that the chemical stress associated with heavy metals, antibiotics, or both could lead to enhanced resistance of the bacterial community during wastewater treatment. It is worth noting that the heavy metals are not biodegradable and their concentration in wastewater is generally 2–4 fold higher than that of antibiotics [10], which could result in persistent selection for heavy metal resistance [25]. A greater understanding and knowledge of stress factors are crucial for determining potential model genes for determining co-selection scenarios.

The presence of organic matter is also important from the perspective of its interaction with ARGs. Recently, the role of organic carbon in shifting the relative abundance of ARGs was investigated in a sand filter biofilm [26]. A decrease in the concentration of total organic carbon (TOC) was found to increase the diversity and relative abundance of ARGs, suggesting that lower levels of TOC were more favorable for enhanced antibiotic resistance. Although this study was carried out during sand filtration of drinking water, the changes in richness, absolute abundance, and relative abundance of ARGs associated with changing concentrations in TOC could be similar during other wastewater treatment processes. In fact, a higher concentration of organic matter in wastewater compared with drinking water could potentially imply a greater richness of the bacterial community, which might lead to a greater increase in the diversity and relative abundance of ARGs upon the gradual decrease in organic carbon during different wastewater treatment processes. The authors also analyzed the changes in the antibiotic resistome with the depth of the sand filter and found that the relative abundance of ARGs increased with the depth and richness of ARGs, correlating positively with the respective TOC levels [26]. In addition to a reduced concentration of organic carbon, the oligotrophic environment was also found to be favorable for the growth and survival of ARBs.

There is a whole range of factors that could change the antibiotic resistome during various stages in a WWTP. The type of wastewater, concentration of organic carbon, and treatment train applied affect the level of ARB and ARGs and resultantly correlations between different genes and mobile genetic elements at a WWTP. It is therefore important to assess the role of each factor in contributing to the final antibiotic resistance scenario considering the impacts related to the discharge or reuse of wastewater.

## 3. UV Radiation for Controlling Antibiotic Resistance

### 3.1. UV Photolysis for DNA Damage

UV photolysis is among the most investigated processes for ARG inactivation at lab-scale, which is largely due to its effective application for conventional disinfection. Several studies have recently investigated inactivation of various classes of ARGs (Table 1). As shown in Table 1, the extent of damage to ARGs varies significantly between different studies. The range of UV doses applied also shows significant variations as discussed later in this section.

The impact of UV irradiation has been investigated for a range of ARGs. For example, changes in the *tet* and *sul* ARGs located on chromosomes were investigated after exposing ARBs to two different UV fluences (5 and 10 mJ/cm^2^) [27]. The impact of UV irradiation was studied for both chromosomal and plasmid DNA in terms of average harboring frequency. The trends were found to be different for both types of DNA. The average harboring frequency of ARGs located on chromosomal DNA was 2–3% for *tet* whereas for *sul* genes it was 14% (*p* < 0.05). These changes were primarily attributed to the changes in the microbial community post-UV irradiation. In addition to changes in the bacterial community, it was postulated that there might be interactions between ARGs and genes related to UV defense (i.e., co-selection), resulting in an increase in chromosomal ARGs. A similar finding was reported in an earlier investigation in which genes related to oxidative stress and protective mechanisms, including cellular repair, were found to be upregulated in multiantibiotic-resistant *E. coli* after solar irradiation [28]. Mechanisms of UV disinfection affecting ARG conjugation and transfer are shown in Figure 2. It shows that UV alone has little effect on the cell membrane but results in damage to the plasmid containing ARGs, resulting in the death of the donor or the recipient [29].

The authors noted that the trends for plasmid DNA were different to that for chromosomal DNA. For example, six ARGs (*tet*A, *tet*C, *tet*M, *tet*W*, tet*X*,* and *sul*1) showed a decrease upon UV irradiation with an average reduction of 15% and 6% for the *tet* and *sul* ARGs, respectively [27]. A 30% reduction in the concentration of bacteria harboring three to five *tet* genes was reported whereas the ratio of bacteria simultaneously carrying both *sul*1 and *sul*2 genes also reduced, although no data were provided on the level of reduction. The authors hypothesized that the reduction in the plasmid ARGs was predominantly caused by the loss of plasmids. Moreover, chromosomal DNA was concluded to be more stable to UV exposure and thus require a greater UV fluence compared with the plasmid DNA. Furthermore, the authors calculated the MAR index according to the following formula [27].
MAR index = a/(b × c)
where a is the aggregate resistance score of all isolates from one sample (if one isolate was observed to resist one antibiotic, it will obtain one point; if not, it will obtain zero points), b is the number of antibiotics used, and c is the number of isolates.

The MAR index was determined for phenotypes of cultivable isolates that were resistant to ten antibiotics (tetracycline, sulfadiazine, cephalexin, penicillin, erythromycin, vancomycin, gentamicin, chloramphenicol, ofloxacin, and ciprofloxacin [27]). The authors found that the MAR index increased markedly at a lower UV fluence (5 mJ/cm^2^), whereas increasing the UV fluence to 10 mJ/cm^2^ did not lead to any changes in the MAR index. It was related to the significant changes in the bacterial community at a higher UV fluence with genera having a lower MAR index dominating at the higher UV fluence.

Acknowledging the significance of UV fluence for different ARGs, the effectiveness of UV irradiation in inactivating ARGs (*bla*_TEM1_, *tet*A, *sul,* and *mph*A) was investigated at various UV fluences by Destiani and Templeton [30]. The lowest level of inactivation was noted for the *mph*(A) gene with 0.05-log inactivation at 20 mJ/cm^2^ increasing to 0.42-log at the highest UV fluence (200 mJ/cm^2^) used in their study. The *tet*(A) gene showed a slightly greater level of inactivation than that for the *mph*(A) gene with 0.05-, 0.36-, 0.38-, and 0.74-log inactivation at UV fluences of 20, 50, 100, and 200 mJ/cm^2^, respectively. The *bla_-_*_TEM1_ gene was the least resistant to UV irradiation with 1.18-log inactivation at a UV fluence of 200 mJ/cm^2^. Hence, the order of susceptibility of ARGs was *bla*_TEM1_ ≥ *tet*A ≥ *sul*1 ≥ *mph*A, essentially following the order of potential dimers, specifically TT dimers. The number of thymine dimer sites has previously been correlated with UV susceptibility, i.e., the greater the number of sites the greater the UV damage [30]. For example, *amp*C, which has a lower number of thymine dimer sites, was reported to be much more resistant to UV damage compared with *mec*A with a higher number of thymine sites. Furthermore, these findings agree with another investigation in which a comparable level of inactivation of both ampicillin and kanamycin ARGs was correlated to the comparable number of thymine sites [31]. Additionally, it must be noted that the actual damage to ARGs could be higher depending on the amplicon length (which the authors did not report) used in the qPCR assay. It has recently been shown that qPCR could underestimate the damage to ARGs [32], particularly for short amplicons.

The impact of UV irradiation on ampicillin-resistant *E. coli* (CGMCC 1.1595) as well as the plasmid-encoding ampicillin resistance gene *bla_TEM-1_* was assessed after exposure to different UV fluences [33]. A range of UV fluences was tested and the damage to ARG was found to be 0.5-log at a UV fluence of 20 mJ/cm^2^, whereas the inactivation ratio of viable *E. coli* was >2.0-log at a similar UV fluence. The ARG damage started to become more prominent with increasing UV fluence such that it increased to 1.2-log at 80 mJ/cm^2^. The inactivation ratio of viable *E. coli* was much higher (6-log) at half the UV fluence (40 mJ/cm^2^), demonstrating that the extent of *E. coli* inactivation was significantly higher than the ARG damage.

A relatively higher UV fluence value of 600 mJ/cm^2^ was used in another study [34]. The authors used drinking water and various ARGs in *E. faecium* and *E. coli*. Despite a 3-fold increase in the UV fluence compared with Destiani and Templeton [30], the authors reported a lower reduction (0.33-log) in the gene copy number of various ARGs (*erm*B, *van*A) using *E. faecium*. When using *E. coli*, they reported no reduction in the tetracycline resistance gene (*tet*A), whereas the reduction in the *β*-lactam resistance gene (*amp*C) was 1-log. Furthermore, the authors looked at the removal of selected ARGs at two different amplicon lengths, i.e., *tet*A (160 and 1054 bp) and *van*A (377 and 1030 bp). In agreement with others [32,35], a greater log reduction (up to 1-log) in *tet*A and *van*A was reported when using long-amplicon qPCR compared with lower amplicon lengths, for which the reduction was negligible.

A much higher UV fluence of 12,477 mJ/cm^2^ was reported to obtain ~2.5-log damage of four ARGs (*sul*1, *tet*G, *intI*1, and 16S rDNA) in wastewater containing COD of 13–29 mg/L [36]. The authors reported that the damage to 16S rDNA and *intl1* flattened with increasing UV fluence from 1248 to 3743 mJ/cm^2^, whereas other ARGs showed greater damage with increasing UV fluence. No explanation was provided as to why increasing the UV fluence did not result in increased damage to 16S rDNA and *intl1*. The authors also noted an increase in the relative abundance (the gene copy numbers of ARGs and *intI*1 normalized to that of 16S rDNA) under a UV fluence of up to 1248 mJ/cm^2^. It could be attributed to the corresponding changes in the 16S rDNA upon UV exposure. Contrary to the damage to ARGs reported by Zhuang et al. [34], no reduction in *tetG* and *tetQ* was reported even after subjecting the wastewater to a UV fluence of 30,100 mJ/cm^2^ [37].

Most of the studies thus far have been conducted at lab-scale, with a very few at full-scale [18,36]. One of the full-scale studies investigated different UV irradiation times from 4 to 18 sec (UV fluence not provided) [36]. As expected, increased damage to DNA and to 16S rRNA and ARGs was reported. However, the overall damage as determined by qPCR was <1-log [33]. These results corroborate previous studies including full-scale applications regarding the limited efficacy of UV photolysis due to quenching by organic materials in wastewater [17]. At a UV fluence of 27 mJ/cm^2^, no significant change in the concentration of ARGs (*tet*X, *tet*M, *tet*A, *sul*1, *sul*2, *erm*B, *qnr*D, and *bla*_TEM_) was found as determined by their relative abundance (ARG copies/16S rRNA gene copies) [18]. Although both these investigations reported a negligible effect on the level of ARGs upon UV irradiation, it is difficult to directly compare the effectiveness of the UV process since it is not possible to compare the UV fluence values between these studies. It highlights the need to report the UV fluence in a standardized unit (mJ/cm^2^) to enable comparative assessment of the process performance. Although UV fluence data were not provided by Rodríguez-Chueca et al. [38], the time of UV irradiation is quite short and hence the results are not surprising. Similarly, the UV fluence used by Lee et al. [18] is also not very high (27 mJ/cm^2^). In another investigation, UV disinfection post-activated sludge treatment at a WWTP in Tunis was reported to be ineffective in reducing the abundance of ARGs [19]. This was not unexpected, since the current UV fluences applied in practice rarely exceed 40 mJ/cm^2^, which is not high enough to damage ARGs.

Although a direct comparison with regard to the ARG damage cf. UV fluence is not possible, it is plausible to argue that the differences in the level of ARG inactivation in different studies could be due to the type of strain (single vs. mixed) and the differences in the characteristics of wastewater. For example, McKinney and Pruden [35] showed an insignificant difference in the UV inactivation of ARGs in two different matrices, i.e., phosphate buffer and wastewater effluent containing TOC of 4.61 mg/L. However, the UV fluence needed to achieve a 3–4-log inactivation of ARGs (*mec*A, *van*A, *tet*A, *amp*C) was much higher (200–400 mJ/cm^2^) compared with the <150 mJ/cm^2^ reported by Yoon et al. [31]. Furthermore, it must be noted that the impact of the matrix could not be truly determined from the findings of McKinney and Pruden [35] since the samples were pre-filtered for turbidity removal. These findings demonstrate that while UV radiation can potentially be useful for damaging ARGs in real-water matrices containing organics and inorganics, the energy and cost associated with these processes are major factors that need to be considered. None of the studies have so far looked at the electrical energy needed for various inactivation/reduction levels of ARGs.

### 3.2. UV Radiation and Chlorine for DNA Damage

The use of chlorine both individually and in combination with UV radiation has been investigated for inactivation of ARGs. Chlorine alone has been reported to be ineffective in damaging most ARGs, except for some, as discussed later in this section. UV radiation can be combined with chlorine to achieve greater damage to microorganisms. UV photolysis of chlorine produces a wide range of highly reactive species, such as HO^•^ and Cl^•^ (a redox potential of 2.4 V) [39,40]. However, it is known that HO^•^ is >5-fold the concentration of Cl^•^ and thus contributes more to the disinfection efficiency [41]. The photodecay rates during the UV/chlorine AOP are related to the wavelength-dependent molar absorption coefficient [42]. The photodecay rate of chlorine has also been shown to increase with increasing pH at any wavelength. Therefore, the pH has to be controlled when using HOCl in the UV/chlorine AOP because it significantly affects the molar absorption coefficient [43]. Mechanisms of chlorination disinfection affecting ARG conjugation and transfer are shown in Figure 3. Contrary to UV radiation alone, which results in damage to the plasmid containing ARGs, HClO reacts first with NH^4+^, leading to the generation of chloramine (mainly NH_2_Cl) that results in cell permeability and ARG transfer. Consequently, chlorination could amplify the risk of ARG transfer, particularly in wastewater with a high concentration of ammonia nitrogen [29].

A sequential UV/chlorine process was investigated for *sul*1, *tet*X, *tet*G, *intI*1, and 16S rRNA genes in municipal wastewater effluent [36]. Compared with UV radiation alone, the sequential UV/chlorine process was found to achieve synergy that ranged between 0.006 and 0.031-log removals for the investigated genes. The greatest energy was achieved for 16S rRNA, with *tet*X showing the least. The amplicon length used in the study by Zhuang et al. [36] was short (163–280 bp), which makes it hard to evaluate the overall gene damage. Furthermore, the concentration of chlorine used was 25 mg/L, which is much higher than the concentration used in practice, which rarely exceeds 2 mg/L [44]. A summary of the literature studies focusing on chlorine alone and the UV/chlorine process is given in Table 2. A range of ARGs have been investigated under different UV radiation and chlorine dose conditions with a fairly high log removal value (LRV) as shown in Table 2.

Most studies have looked at the combined UV and chlorine process with some evaluating the efficiency of chlorine alone for comparison with the combined process. A recent study investigated the reduction of *sul*1 and *intI*1 within Pseudomonas HLS-6, a multiple-antibiotic-resistant bacterium [45]. While the UV/chlorine process showed a greater amount of gene damage in the first 20 min of treatment when compared with chlorination alone, the final gene damage efficiency of *sul*1 and *intI*1 was comparable for both processes. Moreover, both chlorination alone and UV/chlorine treatment gave higher removal efficiencies for both genes (*sul*1, >3.50 log; *intI*1, >4.00 log). The efficiency of damage to the genes was also analyzed by gel electrophoresis, which confirmed the benefit of UV/chlorine treatment on DNA damage. For example, the band intensity of *sul*1 for the UV/chlorine sample treated for 60 min was much lower than that of the samples treated by UV or chlorine alone under comparable conditions. It was further shown that ∼10^2^ copies/mL of *intI*1 remained after UV/chlorine treatment for the shorter amplicon (146 bp) but, for the larger amplicon (484 bp), the gene was found to be completely damaged since no band appeared after 30 min of disinfection [45]. It is therefore plausible to conclude that the disinfection process would be much more effective if the complete sequence size were to be examined for damage detection. Under the condition of a low chlorine dosage, *sul*1 was easier to damage than *intI*1 by the UV/chlorine process. The log damage to both *sul*1 and *intI*1 decreased with increasing pH during the UV/chlorine treatment.

Furthermore, the authors investigated the damage to *sul*1 and *intI*1 under different chlorine concentrations (0–40 mg/L) [45]. A clear increase in the damage to *sul*1 was observed when the chlorine concentration increased to 5 mg/L, but no improvement was seen with a further increase in the dose of chlorine. For *intI*1, however, an increase in the chlorine dose resulted in increasing damage up to 20 mg/L with no further damage observed when the dose was doubled (i.e., 40 mg/L). These results demonstrate that *sul*1 has lower chlorine dose requirements compared with *intI*1, which could be associated with the larger size of *sul*1. Notably, the final log inactivation for *sul*1 and *intI*1 was fairly similar (~4 log) when 5 and 20 mg/L chlorine were used in the UV/chlorine process. It is worth emphasizing that optimizing the chlorine dose is not only important with respect to minimizing the use of chemicals to avoid the formation of disinfection byproducts (DBPs) but also to minimize the self-scavenging of radicals (HO^•^, Cl^•^) under high chlorine dose conditions [46]. Another investigation looked at the damage to the plasmid-encoding ampicillin resistance gene *bla_TEM-1_* after chlorination alone [33]. The authors reported no damage to *bla_TEM-1_* at a chlorine dose of up to 10 mg Cl_2_/L. However, they did not look at the higher chlorine doses that many other authors have investigated to investigate the impact of higher chlorine concentrations on ARG damage.

The impact of pH on the UV/chlorine process was also investigated by Zhang et al. [45]. The damage to both genes decreased with an increase in pH from 5 to 9. Since the quantum yields of HOCl and OCl^−^ were same the during UV/chlorine process, the pH variation was not deemed to affect the formation of HO^•^ and Cl^•^. However, since the consumption of HO^•^ and Cl^•^ by OCl^-^ was several folds faster compared with HOCl, the damage to both genes reduced at a high pH [45]. Moreover, the concentration of HOCl, which has a higher oxidizing capability compared with OCl^−^, was greater at a low pH, leading to higher gene damage. The authors further investigated the role of free radicals (HO^•^) and reactive free chlorine species (Cl^•^, ClO^•^, and Cl_2_^•^^−^) generated during the UV/chlorine treatment [45]. Nitrobenzene (NB) was used a scavenger of HO^•^ (k_HO_^•^-_NB_ = 3.9 × 10^9^ M^−1^ s^−1^). In agreement with an earlier study [43], the authors found no influence of HO^•^ on *sul*1 and *intI*1 damage since the level of inactivation was similar before and after the addition of NB in the UV/chlorine process. The reduction in both genes, however, was greater for the UV/chlorine process than for the UV + chlorine process (k_UV/Cl2_ ≈ k_UV/Cl2+NB_ > k_UV+Cl2_), indicating the role of RCS radials in damaging genes. However, these results differ from the findings of Rattanakul and Oguma [47], who investigated the damage to the viral genome using the UV/chlorine process and found that HO^•^ did result in damage to the genome. The difference in the findings could be attributed to the level of HO^•^ generated during the process and other operational conditions.

The use of different target genes (intracellular bacterial genes vs. the viral genome) in the above-mentioned studies could also be contributing to the different findings. For example, bacteria could consume HO^•^ quicker than bacteriophages due to their complex cellular matrix, minimizing their impact on damaging the genes within bacteria [45]. It can therefore be concluded that the HO^•^ does damage the genes but its impact is dependent on the process’s efficiency and the experimental conditions. In fact, in a recent investigation, it was found that HO^•^ exhibited a very high nonselective reactivity (k ∼10^9^–10^10^ M^−1^s^−1^) towards all nucleobases [48]. The damage to DNA by HO^•^ predominantly occurs by strand fragmentation (via phosphate backbone cleavage) that leads to a reduction in the length of DNA [49]. This ultimately results in a weakened attachment and/or donor–acceptor complexation if exposure to HO^•^ is extended.

Destiani and Templeton [30] found a synergistic effect for inactivating ARGs using sequential UV and chlorine compared with individual treatments. As expected, and in agreement with others [32], the log inactivation of ARGs increased with increasing UV fluence. Sequential UV and chlorine resulted in a synergistic inactivation of the target ARGs [30]. Synergy in the inactivation of the *sul*1 gene was noted for all the ARGs, although some differences were observed. For example, synergy in the inactivation of the *bla*_TEM-1_ gene was observed at UV fluences of 50 and 200 mJ/cm^2^, whereas the synergy for *tet*A was only observed at a UV fluence of 100 mJ/cm^2^ and at UV fluences of 100 and 200 mJ/cm^2^ for *mph*A. The extent of synergy was also dependent on the chlorine dose, with the maximum synergy of 0.25-log occurring for *mph*A at a chlorine dose of 1 mg/L. Increasing the chlorine dose to 2 mg/L (a UV fluence of 200 mJ/cm^2^) resulted in an increased synergy with a 0.6-log greater inactivation of *mph*A compared with the sum of the individual treatments with a final inactivation of 2.8-log. Since free chlorine predominantly reacts with amino acids and membrane-bound proteins and UV radiation with nucleic acids, the synergy in the ARG inactivation during the sequential UV and chlorination process could be attributed to the decrease in bioactivity due to UV irradiation resulting in enhanced reaction of chlorine with cells [11]. The authors also investigated the inactivation of ARBs and found that the required UV fluence was much higher for ARGs than for ARB. Using a chlorine concentration of 30 mg/L, the inactivation of *tet*A, *bla*_TEM-1_, *sul*1, and *mph*A was 1.7-log, which was higher than using UV irradiation alone at a fluence of 200 mJ/cm^2^ (1.2-log).

It could be concluded that a pre-disinfection step (e.g., ozone or chlorine dioxide) leading to substantial cell envelope damage followed by a downstream post-disinfection step using an oxidant for DNA damage might lead to a synergistic effect [50]. Hence, it is plausible to hypothesize that the synergy could be attributed to multiple mechanisms related to two different disinfections. For example, dual damage mechanisms in which chlorine inflicts damage to cell walls and UV irradiation to purines, pyrimidines, and nucleic acid could result in synergistic ARG inactivation [11].

Another recent investigation looked at UV/chlorine synergy for bacterial inactivation and ARG damage [51]. Using chlorine alone (2 mg/L), the copy number of 16S rRNA genes in *Morganella morganii* and *Enterococcus faecalis* decreased by 4.19-log and 3.99-log, respectively. When UV radiation (320 mJ/cm^2^) was combined with chlorine (2 mg/L), the reduction in the copy number of ARGs increased up to 1.5-log. The authors further noted that the inactivation of the *tet*B gene in *Enterococcus faecalis* was greater than that in *Morganella morganii* under similar conditions (*p* < 0.05). In agreement with Destiani and Templeton [30], ARG inactivation needed higher UV fluences compared with ARB. Overall, it was concluded that the inactivation of ARGs was more pronounced at a higher chlorine concentration and was of greater significance than a higher UV fluence. The effect was further enhanced when UV and chlorine were combined. For example, at a UV fluence of 40 mJ/cm^2^, the reduction in ARGs ranged between 0.15 and 0.38-log, whereas adding 2 mg/L chlorine increased it to 0.41–0.94-log.

Combining UV radiation with chlorine is an emerging AOP that potentially could be retrofitted in most water and wastewater facilities. Compared with stand-alone UV and UV/H_2_O_2_ treatments, UV/chlorine treatment could lead to minimizing the selection of resistant genes by reducing chlorine requirements, which would also be beneficial in lowering the formation of DBPs. Simultaneously, the UV fluence could also be reduced when chlorine is used in the UV/chlorine AOP, which would reduce the energy requirements and hence improve the economic efficiency of the combined treatment. However, research in this area is quite limited and further studies looking at the optimization of the process are needed. Further work to determine the role of different conditions (such as the radical generating agent, its concentration in a UV/chlorine AOP, the impact of gene-carrying organisms (bacteriophages or bacteria), and the nature of genes (intracellular or extracellular)) is important in determining the impact of oxidizing species, particularly HO^•^. Additionally, more research optimizing the chlorine dose in relation to UV fluence would be useful to understand and enhance the synergy for ARG inactivation. Nonetheless, the UV/chlorine AOP could be a promising practical alternative to not only improve the inactivation of ARGs and ARB, but to also simultaneously reduce the possibility of bacterial regrowth in water distribution systems as well as minimize microbial selection and the formation of DBPs [2].

### 3.3. UV/H_2_O_2_ AOP for Damaging DNA

The UV/H_2_O_2_ process relies on photolysis of H_2_O_2_ to generate HO^•^ and is among the most widely investigated and applied AOPs in water and wastewater treatment at laboratory and/or small scale with huge potential for full-scale applications. The process has been increasingly used for damaging ARGs in recent years with a few full-scale investigations (Table 3). A recent full-scale study employed the UV/H_2_O_2_ process for damaging different ARGs in tertiary treated wastewater, i.e., wastewater post-coagulation/flocculation/decantation followed by filtration by a rotofilter [36]. The UV/H_2_O_2_ process was the most efficient in damaging 16S rRNA and other investigated ARGs when compared with other AOPs including UV/peroxymonosulfate (PMS) with or without Fe (II) [36]. Although the generation of radicals was not investigated by the authors, it was hypothesized to be higher during UV/PMS-based processes compared with the UV/H_2_O_2_ process. It must, however, be noted that the species of radicals produced are different during these processes, with HO^•^ being non-selective in nature compared with the selective S$O_{4}^{•-}$ generated during UV irradiation of PMS. It is also worth noting that the redox potential of PMS (+1.82 V) is higher than that of H_2_O_2_ (+1.76 V) [52]. Some lab-scale investigations have demonstrated UV/PMS to be superior to the UV/H_2_O_2_ process [53,54]. UV/PMS is outside the scope of this review but it is apparent that the efficiency of UV/PMS is more dependent on the type of water matrix when compared with UV/H_2_O_2_ [55]. Further work is therefore needed to fully understand the UV/PMS process and its robustness with respect to damaging ARGs, particularly in representative water matrices.

The impact of HO^•^ on the inactivation of ARGs is one of the most focused-on aspects of most studies investigating the UV/H_2_O_2_ process, with varying findings. For example, Yoon et al. [31] reported a negligible contribution of HO^•^ to the inactivation of e-ARGs (*amp* and *kan*) during UV/H_2_O_2_ treatment of wastewater effluent. It must be noted that the wastewater effluent samples were collected from the conventional activated sludge process and therefore scavenging of radicals was expected. According to the authors, the average UV fluence delivered in the case of wastewater was 1.4-fold lower than the UV-transparent water, such as phosphate buffer solution. The rates of e-ARG damage were fairly similar for the UV-only and UV/H_2_O_2_ treatments (*p* = 0.56 and 0.75, respectively) due to the reduced HO^•^ oxidation efficiency in the wastewater effluent matrix due to radical scavenging by organic matter (DOC = 5.2 mg/L). Although the authors did not support their results with an actual measurement of radicals, the fact that the e-ARG damage in the phosphate buffer was greater during the UV/H_2_O_2_ treatment compared with the UV treatment corroborates their findings of radical scavenging. Therefore, despite the high HO^•^ reactivity to e-ARGs (*k* = ~10^10^ M^−1^ s^−1^), the resulting degradation of ARGs could be insignificant during UV/H_2_O_2_ treatment of complex water matrices [31].

Another study investigated the impact of a 250 W lamp equipped with a UV filter (emission range: 320–450 nm) in UV/H_2_O_2_ (20 mg/L) for its potential to reduce resistance transfer [56]. The antibiotic-resistant *E. coli* strain was isolated from the effluent of an activated sludge process of a WWTP and cultivated on selective culture medium. Qualitative PCR was performed on total DNA and on DNA extracted from cell cultures to investigate the *bla*_TEM_, *qnr*S, and *tet*W genes in *E. coli*-spiked samples. While the *bla*_TEM_ gene was detected in both samples, *qnr*S and *tet*W were not detected in the PCR assay; they were either absent or present at very low concentrations. The results showed no change in the DNA extracted from cell cultures after UV/H_2_O_2_ treatment for up to 90 min. Similar results were reported for total DNA with the *bla*_TEM_ gene copy number remaining unchanged during treatment for up to 300 min despite total inactivation of *E. coli* after 240 min. Although the treatment time was very high, the UV fluence delivered at this time was quite low (25 mJ/cm^2^) compared with other investigations reporting several hundreds of UV fluence needed for a 2–4 log ARG inactivation. Moreover, the wavelength range used by the authors is not commonly used in real applications. Additionally, the efficiency of the UV/H_2_O_2_ process is highly dependent on the concentration of H_2_O_2_ and other water quality parameters that need to be optimized for achieving effective inactivation.

The impact of operating parameters such as H_2_O_2_ dose and pH has been well reported during the UV/H_2_O_2_ process for conventional applications, i.e., degradation of organics in water and wastewater. Some recent studies have also been carried out to determine their impact on the ARG damage in wastewater. For example, an optimized UV/H_2_O_2_ process taking pH, H_2_O_2_ dose, and time of UV irradiation into account was investigated for various ARGs (*sul*1, *tet*X, *tet*G, *intl*1, and 16S rRNA) in secondary effluent using a 254 nm UV lamp [57]. At a much higher concentration of H_2_O_2_ (340 mg/L) and a pH of 3.5 compared with the study of Ferro et al. [57], a much higher reduction in the investigated genes in the range of 2.64–3.48 after 30 min of irradiation was reported by Zhang et al. [58]. Increasing the pH to 7 resulted in a reduced log reduction of ARGs to 1.55–2.32. Therefore, a pH of 3 and a H_2_O_2_ concentration of 340 mg/L were considered best for damaging ARGs. Such a lower pH is not practically feasible. Therefore, optimization of process parameters for ARG inactivation needs to be carried out specifically considering practical application of the UV/H_2_O_2_ process. As the concentration of H_2_O_2_ is one of the most important factors determining the efficiency of the UV/H_2_O_2_ process, the impact of H_2_O_2_ concentration on the inactivation of two ARGs (*mec*A and *amp*C) has also been investigated by others [59]. For a UV fluence of 120 mJ/cm^2^, approximately 2.3–2.9- and 1.4–2.7-log inactivation of *amp*C and *mec*A, respectively, was achieved with different concentrations (340, 1700, and 3400 mg/L) of H_2_O_2_. With the addition of thin TiO_2_ film, the inactivation of *mec*A and *amp*C improved to 2.7–3.4- and 2.7–3.2-log, respectively.

It is clear from the studies reported thus far that the UV/H_2_O_2_ process is an effective measure to damage ARGs. However, the UV fluence and H_2_O_2_ dose required could be very high for a real-water matrix with organics. The focus therefore needs to be placed on appropriate pre-treatment technologies to reduce the organic load to minimize costs and enhance process efficiency. Considerable research has been carried out to determine the efficacy of different pre-treatments prior to UV AOPs for the removal of contaminants to improve the subsequent process performance, which could make an excellent starting point for investigating the ARG damage in real matrices by combining UV/H_2_O_2_ with suitable pre-treatments. A biological treatment as a post-UV/H_2_O_2_ treatment is generally very effective in improving the energy efficiency and overall treatment performance. It is, however, uncertain how the combined UV/H_2_O_2_ and biological process performs with respect to changes in ARGs, which needs to be investigated in future studies. UV-LEDs as emerging sources of UV irradiation could prove more effective considering the possibility of combining different wavelengths in novel reactor designs. Only one study has thus far looked at the application of UVC-LEDs for damaging ARGs [60].

A shared concern with most disinfection processes as well as UV-based processes is the selection of ARGs that could result in transfer of antibiotic resistance even after treatment. The focus of future studies irrespective of the UV source needs to be placed on determining the optimum conditions to minimize or avoid selection pressure. Within this context, benchmarking UV fluences and the role of HO^•^ in combination with other operational conditions should be considered in future investigations with a view to reducing the overall ability of ARGs to transfer resistance.

## 4. Current and Future Perspectives

Considering that antimicrobial resistance is a recent concern when compared with traditional contaminants in water and wastewater, there are several unknowns and uncertainties that need to be understood to enable the control or mitigation of antibiotic resistance. One of the greatest challenges is to identify practical and relevant target types in addition to levels of antibiotic resistance benchmarks specifically for reuse applications [1], which are important to providing or formulating a basis for potential future regulations. Moreover, standardized methods for both the identification and quantification of ARB and ARGs are needed. Since identifying all the ARB and ARGs is not practical, there is a need to identify targets both in terms of the types and final level of inactivation considering the deleterious health impacts—an approach analogous to the one adopted for pathogens as microbiological water quality indicators. Although it is uncertain if such a strategy could be applied to DNA/ARGs, it could be a good starting point in identifying potential approaches to answer some of the key questions related to tackling the challenge of antibiotic resistance.

There is a considerable knowledge gap in relation to the fate of ARB and ARGs during conventional wastewater treatment and the future role of advanced treatment technologies in addressing the important issue of antibiotic resistance. For example, little is known about the impact of potential factors on the selection of and changes in ARB and ARGs during different wastewater treatment stages. Moreover, knowledge on how the overall bacterial community, antibiotics, metals, and other selective agents relate to the abundance or removal of clinically relevant ARB and ARGs under selected treatment schemes is lacking. This information is important to understand the conditions that favor the growth of ARBs and to understand the associated risks. Furthermore, because of DNA repair mechanisms in the cells, it is uncertain how much DNA damage is needed to make ARGs useless to bacteria, i.e., to inactivate ARGs permanently or make them unsuitable for transfer. Therefore, it is important to establish quantitative DNA assays and relate them to culture-based assays after UV irradiation. Advanced treatment technologies, including UV and membrane filtration, have been successfully implemented for wide-ranging applications in the water industry and these have been shown to be effective for controlling antibiotic resistance. In regard to sources of UV irradiation, UV-LEDs as robust and energy-efficient sources of UV irradiation are of particular interest and need to be investigated in the future.

Wastewater disinfection processes operated under typical treatment conditions are markedly effective in minimizing the overall ARB levels; however, they could lead to ARB selection (i.e., increased relative proportions of ARB amongst the surviving bacterial cells) [61,62]. This could lead to an increase in the potential for antibiotic resistance dissemination. Therefore, processes aimed at achieving disinfection need to shift the focus from microbial inactivation to achieving ARG inactivation. It is important to note that the selection pressure exerted on environmental bacteria depends on several factors, including the type of selective agent, their concentrations and chemical speciation, co-exposure to other selective agents, exposure time, and environmental conditions for bacterial growth [63]. It is therefore critical to understand the complexities and uncertainties involved in these factors to completely recognize the role that selective agents play in promoting the antibacterial resistance [64].

Conventional UV lamp systems are a well-established technology for water disinfection, but most studies conducted thus far have investigated “log inactivation of ARGs”. A higher or a specific log inactivation could be a useful measure for bacterial inactivation, but it does not necessarily reflect the safety of the treated water in terms of the remaining resistance potential. It is therefore important to consider how the log ARG inactivation correlates with the ARG transformation potential after treatment, which requires simultaneous application of both DNA quantification and plate count methods for determining transformation potential. Considering that AOPs can achieve the dual objective of organics degradation, including several antibiotics, and microbial disinfection, including DNA damage, these processes need to be thoroughly investigated under different conditions. This is important to understand the role of different radicals (Cl^•^, Cl_2_^•^^−^, HO^•^) in simultaneously oxidizing organics and damaging DNA to minimize opportunities for antibiotic resistance spread. The effectiveness of AOPs has been proven for the inactivation of ARB and damage to ARGs at lab-scale. However, their application on real wastewater matrices at full-scale is challenging and requires research in a broad range of areas, including the optimization of the processes in the presence of contaminants, the impact of pre- and post-treatments on the process efficiency, and the overall operational cost and energy requirements under different scenarios (oxidant dose, type and quality of wastewater, UV fluence). UV-LEDs could become practically applicable in the future, opening up more opportunities to combine different wavelengths for the purpose of effective microbial and DNA damage.

The challenges that remain to be addressed are multidisciplinary in nature. From initial clinical control of antibiotic administration to their spread in the aquatic environment, questions related to the types of antibiotics and associated ARGs that are of most concern include: What is their removal and bacterial uptake potential? Which ARGs pose the maximum level of risk? How is the maximum level to be defined? How does abundance relate to risk in a wider environmental perspective? All these aspects are important to be considered for an accurate risk assessment and devising comprehensive and economically viable treatment strategies. Since treated wastewater could be reused multiple times and for diverse applications, such as indirect potable and industrial use, irrigation, and aquifer recharge, assessing the relevant risks is even more complex when compared with drinking water. Therefore, the need to adequately address the relationship between anthropogenic and environmental factors in assessing health risks cannot be overemphasized.

## 5. Conclusions

Antimicrobial-resistant bacteria and antimicrobial resistance genes present an emerging challenge to treated wastewater reuse applications considering that these contaminants are unregulated. These concerns are further exacerbated by the fact that: (1) wastewater treatment plants are considered a hotspot of microbial resistance; (2) the fate of antimicrobial resistance genes is not understood; and (3) the current treatment approaches are inefficient in inactivating genes. UV-based processes are some of the most investigated advanced treatments that have been tested and validated at large-scale with many full-scale applications globally. UV and chlorine as stand-alone processes could be effective, but modifications (combined use, use of oxidants, pre-treatments) need to be made for their efficient application. Nonetheless, the efficiency of these processes needs to be investigated with an approach different to what has been prevalent in the water industry over the past few decades, i.e., from inactivating indicator microorganisms to DNA damage. Hence, there is a need for a paradigm shift from conventional disinfection, with the primary aim of inactivating pathogens as an indicator of the microbiological safety of water, to effective damage to the DNA and resistance genes that could still be present after microbial inactivation. This approach requires an understanding of the types of genes and their fate during wastewater treatment, potential health and ecological impacts, the transformation potential, and the impact of conventional treatments cf. advanced treatments both on the inactivation and relative abundance of different types of genes. Considering the limited knowledge on the real impact (on human health, on ecosystems) of the presence of environmental ARB and ARGs present in source water, treatment plants, and distribution systems and upon wastewater reuse, future research needs to place emphasis on understanding and quantifying the risks. Furthermore, it is critical to understand how the technological interventions would impact the overall treatment and economic outlook of wastewater reuse applications.

## Figures and Tables

**Figure 1 ijerph-19-01636-f001:**
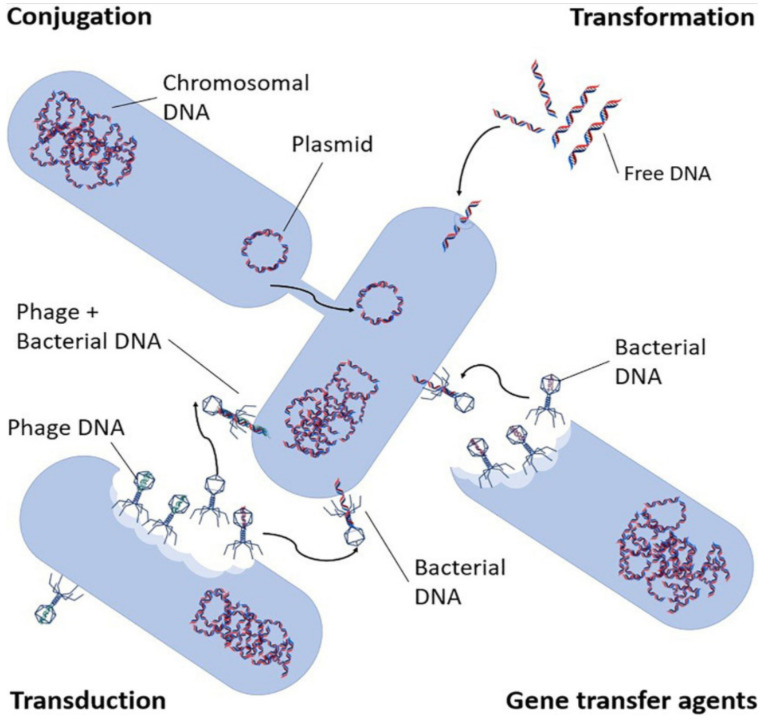
Antibiotic resistance transport mechanisms (adapted from von Wintersdorff et al. [14]).

**Figure 2 ijerph-19-01636-f002:**
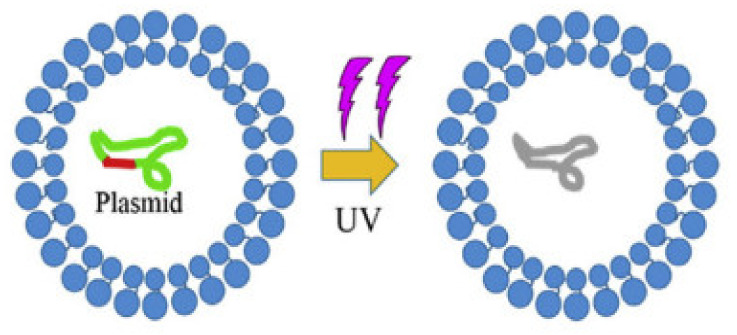
Mechanisms of UV disinfection affecting ARG conjugation and transfer (adapted from Guo et al. [29]).

**Figure 3 ijerph-19-01636-f003:**
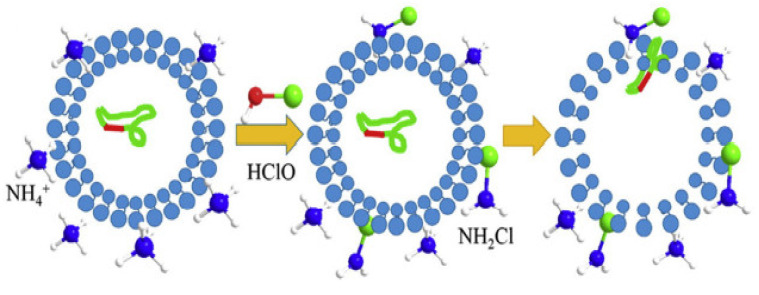
Mechanisms of chlorination disinfection affecting ARG conjugation and transfer (adapted from Guo et al. [29]).

**Table 1 ijerph-19-01636-t001:** Inactivation of ARGs by UV irradiation.

Mode	Peak λ	Volume (mL)/Flow Rate	Target	Log Inactivation	UV Dose (mJ/cm^2^)	*k* (cm^2^/mJ)	Matrix	Reference
Full-scale	NG	130 mgd	*tet*R	0	30,100	NG	WW effluent	[35]
Bench-scale collimated beam	254	10	*mec*A, *van*A, *tet*A, and *amp*C	3–4	200–400	0.4–0.25, 0.015–0.01	PB and WW effluent	[32]
Batch	254	1500	*sul*1, *tet*G, and *intl*1	2.5–2.7	12,477	0.0002	WW effluent	[34]
Batch	254	1800	*sul*1, *tet*X, *tet*G, *intI*1, and 16S rRNA	* <1	62.4, 124.8, 249.5	0.016, 0.008, 0.004	WW effluent	[41]
Full-scale	254	-	*tet*X, *tet*M, *tet*A, *sul*1, *sul*2, *erm*B, *qnr*D, and *bla*TEM	0	27	-	WW effluent	[17]
Lab-scale	254	15	*bla*TEM-1	1.2	80	0.015	PBS	[32]
Collimated beam	254	NG	*bla*_TEM-1_, *tet*A	** 1	20–25	0.05–0.04	Plasmid suspension in DNase-free water	[13]
Bench-scale quasi-collimated beam	254	120	*amp^R^, Kan^R^*	4	60–140	0.11–0.07,0.15–0.09	PB	[29]
Collimated beam system	254	10	*tet*A*, tet*B, *str*B, *sul*2, and *aacC*2	1.6	320	0.005	Hospital WW	[49]
Bench-scalecollimated beam	254	100	*tet*(A), *bla-*_TEM1_, *sul*1, and *mph*(A)	0.42–1.18	200	0.0021–0.0059	PB	[28]
Batch	254		*tet*A, *van*A, and *erm*B	1	600	0.001	DW	[31]

NG, not given; mgd, million gallons per day; * value considered 1; ** 1-log reduction per UV fluence of 20–25 mJ/cm^2^; WW, wastewater; PB, phosphate buffer; DW, drinking water.

**Table 2 ijerph-19-01636-t002:** Inactivation of ARGs by chlorine and UV/chlorine.

Target	Wavelength (nm)	UV Fluence (mJ/cm2)	LRV	Output Power (W)	Cl_2_ Dose (mg/L)	Volume (mL)	Reference
*sul*1, *tet*X, *tet*G, *intI*1, and *16S rRNA*	254	62.4, 124.8, 249.5	2	16	30	1800	[41]
*sul*1, *intI*1	254	120	~3.5–4	NG	20	50	[43]
*tet*(A), *bla-*_TEM1_, *sul*1, and *mph*(A)	254	200	2.2–2.8	NG	30	100	[28]
*tet*A*, tet*B, *str*B, *sul*2, *aacC*2	254	320	2.7–3.1	NG	2	10	[49]
*sulI,* and *intI1*	-	-	* 1.5–2.4	-	20	50	[44]
*bla* _TEM-1_	-	-	* 0	-	10	15	[32]
*tet*A, *bla-*_TEM1_, *sul*1, and *mph*A	-	-	* 3.4–3.6	-	5	200	[29]

* Chlorine alone without UV irradiation.

**Table 3 ijerph-19-01636-t003:** Inactivation of ARGs by UV/H_2_O_2_.

Target	Wavelength (nm)	UV Fluence	LRV	Output Power (W)	H_2_O_2_ Dose (mg/L)	COD (mg/L)	Volume (mL)	Reference
16S rRNA, *sul*1, *sul*2, *qnr*S, *bla*_TEM,_ *bla*_OXA-A_, and *intl*1	254	40–170 J/L	<1	330 W	17	27 ± 3	140,000	[33]
*sul*1, *tet*X, *tet*G, *intI*1, and 16s rRNA	254	NG	2.8–3.5	16	340	13–39	1800 mL	[55]
*bla_TEM_*	320–450	25 mJ/cm^2^	0	250	20	NG	500 mL	[53,54]
*amp*R, *kan*R	254	44–140 mJ/cm^2^	4	NG	10	(DOC, 5.2)	120 mL	[30]

## Data Availability

Not applicable.

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
