# Peer review of "From Conventional Disinfection to Antibiotic Resistance Control—Status of the Use of Chlorine and UV Irradiation during Wastewater Treatment"

_ijerph, 2022, doi:10.3390/ijerph19031636_

Round 1
Reviewer 1 Report
Only a minor detail, despite the fact that the article is well structured, I suggest to the author to use more diagrams/figures if not, improve the quality of the existing ones.
Author Response
We thank the reviewer for taking time and providing highly valuable feedback. Our responses to reviewer’s comments are highlighted in blue in the response file. The changes made in the revised marked manuscript file are also highlighted in blue.

Reviewer 2 Report
The manuscript describes the use of different disinfection techniques applied in wastewater treatment plants for the removal of ARB and ARG. Although this is a very relevant topic, being in the top 10 list of the WHO, the manuscript itself is at times confusing and difficult to read. Therefore, prior to publication the author should revise the manuscript and arrange it in a more readable format. The author also states that hydrogen peroxide is extensively used in wastewater treatment plants. To the best of my knowledge, this technique is not extensively used and if so, this sentence requires references. Additionally, the figures need some explanation as the legends or the text says very little about them, the author adds abbreviations without writing them in full first (please check this) and throughout the text, several typos exist (check for instance, concentration).
Author Response

(The authors gave the same response as above.)

Reviewer 3 Report
The work is a review of 63 literature items. The topic of work is important to human health and the environment.
Wastewater reuse has traditionally been considered safe after treatment by a combi-nation of physicochemical and biological treatments. However, microbial inactivation alone is not sufficient for ensuring safety of treated wastewater since genes may still be present, even after dis-infection. Importantly, the current disinfection processes are not designed to damage genes which make antibiotic resistance genes (ARGs) a serious concern since they enable micro-organisms to become resistant to antibiotics.
Municipal wastewater is regarded as one of the main sources of antibiotic resistant bacteria (ARB) and ARGs in aquatic environment. That is why the work on this topic are very important and should be of great interest to the readers.
The research is interesting and original, the work is written correctly, the text is clear and easy to read, the layout is logical and the conclusions are correct. There is no unnecessary self-citation at work. The work can be published
Author Response

(The authors gave the same response as above.)
